# GO/Bi_2_S_3_ Doped PVDF/TPU Nanofiber Membrane with Enhanced Photothermal Performance

**DOI:** 10.3390/ijms21124224

**Published:** 2020-06-13

**Authors:** Wenxiu Yang, Yonggui Li, Long Feng, Yimiao Hou, Shuo Wang, Bo Yang, Xuemin Hu, Wei Zhang, Seeram Ramakrishna

**Affiliations:** 1College of Textile and Garments, Hebei Province Textile and Garment Technology Innovation Center, Hebei University of Science and Technology, Shijiazhuang 050018, China; wenxiuyang-hbust@outlook.com (W.Y.); fenglongxxl@outlook.com (L.F.); fzwangshuo@hebust.edu.cn (S.W.); yangbo564@outlook.com (B.Y.); bobzhang69@outlook.com (W.Z.); 2Fujian Key Laboratory of Novel Functional Textile Fibers and Materials, Minjiang University, Fuzhou 350108, China; LiYonggui@mju.edu.cn; 3School of Environmental Science and Technology, Hebei University of Science and Technology, Shijiazhuang 050018, China; ws15222329526@outlook.com; 4National Joint Local Engineering Research Center for Volatile Organic Compounds and Odorous Pollution Control, Shijiazhuang 050018, China; 5Center for Nanofibers & Nanotechnology, Nanoscience & Nanotechnology Initiative, Faculty of Engineering, National University of Singapore, Singapore 117576, Singapore

**Keywords:** graphene oxide/bismuth sulfide, solar distillation, nanofiber membrane, photothermal conversion

## Abstract

Photothermal conversion materials have attracted wide attention due to their efficient utilization of light energy. In this study, a (GO)/Bi_2_S_3_-PVDF/TPU composite nanofiber membrane was systematically developed, comprising GO/Bi_2_S_3_ nanoparticles (NPs) as a photothermal conversion component and PVDF/TPU composite nanofibers as the substrate. The GO/Bi_2_S_3_ NPs were synthesized in a one-step way and the PVDF/TPU nanofibers were obtained from a uniformly mixed co-solution by electrospinning. GO nanoparticles with excellent solar harvesting endow the GO/Bi_2_S_3_-PVDF/TPU membrane with favorable photothermal conversion. In addition, the introduction of Bi_2_S_3_ NPs further enhances the broadband absorption and photothermal conversion properties of the GO/Bi_2_S_3_-PVDF/TPU composite membrane due to its perfect broadband absorption performance and coordination with GO. Finally, the results show that the GO/Bi_2_S_3_-PVDF/TPU composite membrane has the highest light absorption rate (about 95%) in the wavelength range of 400–2500 nm. In the 300 s irradiation process, the temperature changes in the GO/Bi_2_S_3_-PVDF/TPU composite membrane were the most significant and rapid, and the equilibrium temperature of the same irradiation time was 81 °C. Due to the presence of TPU, the mechanical strength of the composite film was enhanced, which is beneficial for its operational performance. Besides this, the morphology, composition, and thermal property of the membranes were evaluated by corresponding test methods.

## 1. Introduction

Photothermal conversion is an emerging approach using light energy which can be widely used in the fields of heat therapy, domestic water heating, photocatalysis, and others [1,2,3]. However, low energy density is a problem to be overcome in light energy utilization technology. How to improve the efficiency of energy conversion is an urgent problem. Among several technologies for light energy utilization (including photothermal, photochemical, and photovoltaic conversion), photothermal conversion has the highest energy conversion efficiency [4].

So far, photothermal conversion materials are generally divided into the following three categories: (1) metal nanoparticles/metal oxides, (2) carbon-based materials, (3) polymer materials [4]. Xu et al. [5] used a CuO nanowire mesh as a photothermal conversion material. Under sun illumination, the solar absorption was 93%. Besides this, the CuO nanowires achieved superior photocatalytic and antibacterial properties. Ren et al. [6] reported that a hierarchical graphene foam was used as a solar harvesting material by chemical vapor deposition technology. Compared to conventional graphene, hierarchical graphene endowed the solar distillation system with considerable solar thermal and solar vapor efficiencies of 93.4% and 91.4%, respectively. Hao et al. [7] used coated cotton fabrics with polydopamine and polypyrrole; the resulting photothermal conversion system was up to 95% absorbent over wavelengths ranging from 200 nm to 1100 nm. Xu et al. [8] developed a core-shell structured polydopamine/polyethyleneimine/polypyrrole @polyamide membrane (PDA/PEI/PPy@PI) by electrospinning. The resulting composite not only possessed superior light absorption (around 93%) but also exhibited a great corrosion resistance. Compared with polymer materials, metal nano-ions/metal oxides and carbon-based materials can make the temperature rise faster under the same light conditions. The local highly sensitive thermal effects have made them ideal photothermal conversion agents in solar harvesting. Carbon-based materials are more environmentally friendly and have better broad band absorption and chemical stability compared with metal nanometer ions/metal oxides [9,10].

Recent studies have shown that bismuth sulfide nanoparticles (Bi_2_S_3_ NPs) possess high cost performance, outstanding biocompatibility, and the great absorption of near-infrared and visible light (UV-NIR) [11,12,13]. The band gap width of the Bi_2_S_3_ NPs is 1.3–1.7 ev, which makes the Bi_2_S_3_ NPs have a high near-infrared absorption coefficient [14,15]. Under the action of visible or near-infrared (NIR) light, Bi_2_S_3_ NPs can be excited to generate heat and have a high conversion efficiency [16,17]. Although bismuth sulfide (Bi_2_S_3_) is widely used in electricity, optics, electrochemistry, biology, and chemo-photothermal therapy for cancer, Bi_2_S_3_ NPs being applied as a photothermal conversion material is seldom discussed. Unfortunately, the low specific surface area of Bi_2_S_3_ NPs leads to its poor solar harvesting performance. However, Bi_2_S_3_ NPs are difficult to spin with polymer materials because they tend to aggregate and have poor dispersion in electrospun solutions. Graphene oxide (GO) has a fantastic large specific surface area and is capable of loading various nanoparticles. Therefore, by loading Bi_2_S_3_ NPs into GO, the resulting GO/Bi_2_S_3_ composite nanoparticles (GO/Bi_2_S_3_ NPs) are an ideal solar harvesting and photothermal conversion agent. Polyvinylidene fluoride (PVDF) nanofiber membrane is widely used in lithium battery separators, air filtration, sensors, and other fields because of its superior chemical stability, high specific surface area, and high porosity. However, the mechanical performance of the PVDF nanofiber membrane is unsatisfactory. Meanwhile, thermoplastic polyurethane, with strong polar groups, excellent mechanical properties, and ageing resistance, has been widely used in many fields requiring high mechanical properties.

In this study, a one-step way was employed to prepare GO/Bi_2_S_3_ NPs as a photothermal conversion material. Considering the requirement of mechanical strength for application, PVDF and TPU electrospun solutions were blended and prepared as the substrate of the photothermal membrane. The PVDF/TPU composite nanofiber membrane can thoroughly combine the advantages of the two polymers. The prepared GO/Bi_2_S_3_ NPs were added to the PVDF/TPU mixed spinning solution to prepare GO/Bi_2_S_3_-PVDF/TPU composite nanofibers by electrospinning. The composite nanofiber membrane was expected to possess excellent mechanical properties and absorbance and be used for cancer photothermal therapy, energy collection, and the distillation of seawater.

## 2. Results and Discussion

### 2.1. Morphology and Composition

A one-step process was used to synthesize GO/Bi_2_S_3_ nanoparticles (NPs). In the synthesis process, Bi(NO_3_)_3_ and thioacetamide (TAA) served as the sources of bismuth and sulfur and polyvinylpyrrolidone (PVP) acted as a surfactant to enhance the stability of GO/Bi_2_S_3_ NPs. Figure 1a shows the XRD patterns of GO/Bi_2_S_3_ NPs. Compared with the standard powder diffraction card (JCPDS 17-0320), the diffraction peaks of Bi_2_S_3_ are observed at 2θ = 22.40°, 24.92°, 28.60°, 31.80°, and 45.66°, which correspond to the (220), (130), (211), (221), and (002) planes of Bi_2_S_3_, respectively. The results indicated that the product produced by hydrothermal synthesis is orthorhombic phase Bi_2_S_3_. The diffraction peak of the GO (100) plane at 10.02° is observed in the GO/Bi_2_S_3_ diffraction pattern, indicating that GO and Bi_2_S_3_ have successfully combined. The XPS elemental survey scan (Figure 1b) shows that the prepared nanoparticles were composed of the C, Bi, S, and O elements. The Bi and S elements correspond to Bi_2_S_3_, and GO is the major source of the C and O elements. The binding energies of Bi 4f7/2, Bi 4f5/2, S 2p1/2, and S 2p3/2 obtained from the corresponding high-resolution XPS spectra (Figure 1c) were located at 158.2, 163.5, 161.2, and 162.4 eV, respectively.

The TEM image (Figure 2a) shows that the Bi_2_S_3_ NPs have a rod-like structure with a diameter range of 10–30 nm and grow uniformly on GO. The lattice fringes with a spacing of 0.39 nm and 0.50 nm correspond to the crystal plane at (130) and (220), respectively. The regular lattice pattern indicates that GO/Bi_2_S_3_ has great crystallinity. HAADF and EDS mapping images (Figure 2b) show the uniform distribution of the elements C, Bi, S and O, which further indicates that the Bi_2_S_3_ NPs can be uniformly dispersed on GO.

Figure 3a shows the TEM image of the GO/Bi_2_S_3_-PVDF/TPU nanofiber (left) and elements mapping of the F, N, Bi, and S (right); the elements mapping image demonstrates the uniform distribution of GO/Bi_2_S_3_ NPs among the PVDF/TPU nanofibers. The SEM images of pristine PVDF and TPU, PVDF/TPU, GO-PVDF/TPU, and GO/Bi_2_S_3_-PVDF/TPU nanofibers are shown in Figure 3b–f, respectively. As shown in Figure 3b–f, the surfaces of the PVDF, TPU, and PVDF/TPU nanofibers are smooth. The surfaces of the GO-PVDF/TPU and GO/Bi_2_S_3_-PVDF/TPU nanofibers are relatively rough. The compositions and structural parameters averages of different samples are exhibited in Table 1. Due to the inherent properties of polymers, the average diameter of the TPU nanofibers is thicker than the PVDF, which results in an uneven thickness of the PVDF/TPU composite nanofibers. Our previous work has shown that the addition of metal oxides and GO to the spinning solution increases the conductivity and reduces the viscosity of the spinning solution, so that nanofibers with thinner diameters can be obtained under the same spinning parameters [18,19]. The results (Table 1) show that the nanofiber diameters of GO-PVDF/TPU and GO/Bi_2_S_3_-PVDF/TPU are smaller than those of PVDF, TPU, and PVDF/TPU, which also supports the conclusion mentioned above.

Table 1 also displays the average pore diameter and porosity of PVDF and TPU, PVDF/TPU, GO-PVDF/TPU, and GO/Bi_2_S_3_-PVDF/TPU membraned. The results manifest that the thinner the diameter of the nanofibers, the smaller the pore diameter but the higher the porosity. The GO-PVDF/TPU and GO/Bi_2_S_3_-PVDF/TPU composite membranes have similar pore sizes of 1.3 and 1.2 μm; both are significantly smaller than the membranes without additives. The porosities of the GO-PVDF/TPU and GO/Bi_2_S_3_-PVDF/TPU composite membranes are the same (83%) and evidently higher than the membranes without additives. These results are still consistent with our previous conclusions [18,19].

Furthermore, the FTIR spectrum was used to verify the formation of the PVDF/TPU, GO-PVDF/TPU, and GO/Bi_2_S_3_-PVDF/TPU composite membranes, as shown in Figure 4. The spectrum of the pristine PVDF and TPU are consistent with literature reports [20,21,22,23,24,25,26,27]. The characteristic absorption peaks of PVDF are observed at 877 cm^−1^, 1174 cm^−1^, and 1402 cm^−1^, which are attributed to the skeletal vibration of the C–C bonds and the stretching vibrations of the C–F and C–H groups, respectively. For the TPU membrane, typical peaks around 1599 and 1716cm^−1^ are observed due to the N-H flexural absorption and stretching vibration of the carbonyl group in the amide, respectively. The absorption peaks at 2943 and 3325 cm^−1^ are attributed to the –CH_2_– asymmetric stretching vibration and the stretching vibration of N–H in the urethane group, respectively. Other characteristic bands of TPU can also be observed in the FTIR spectra of TPU. Compared with the FTIR spectrum of pristine PVDF and TPU, the FTIR spectra of PVDF/TPU contains all the characteristic peaks of both PVDF and TPU, proving the formation of PVDF/TPU composite membranes. In comparison with PVDF/TPU, a new peak appeared at 3428 cm^−1^ caused by the stretching vibration of C=C in GO, illustrating the presence of GO in PVDF and TPU. Other absorption peaks related to GO vanished in the FTIR spectra of GO-PVDF/TPU due to coinciding with the peaks of other components. Moreover, the stretching vibration of Bi–S is observed in the FTIR spectra of the GO/Bi_2_S_3_-PVDF/TPU, indicating that the Bi_2_S_3_ NPs were favorably embedded on the surface of GO [11].

### 2.2. Thermal Property

The thermal behaviors of the PVDF, TPU, PVDF/TPU, GO-PVDF/TPU, and GO/Bi_2_S_3_-PVDF/TPU membranes were evaluated by differential scanning calorimetry (DSC) and TGA, as shown in Figure 5a,b. The DSC curve of PVDF presents an endothermic peak at 169.38 °C, which represents a typical melting peak of the PVDF. The DSC curve of TPU shows two endothermic peaks at 115.66 °C and 283.58 °C, which are caused by the melting of the soft segment and the hard segment of TPU, respectively. On the DSC curve of the PVDF/TPU membrane, there is only one endothermic peak, and the endothermic peak of the soft segment of TPU disappears. With the addition of GO and GO/Bi_2_S_3_ NPs, the melting temperatures of the GO-PVDF/TPU and GO/Bi_2_S_3_-PVDF/TPU composite membranes increased correspondingly.

A TG measurement was used to further characterize the thermal properties of different membranes. As shown in Figure 5b, for PVDF, the weight barely changed until 200 °C. At about 460 °C, the PVDF membrane began to degrade rapidly, and the mass showed a rapid decline. This may be due to the degradation of the carbon skeleton. The residual rate at 800 °C was 10.45%. The pristine TPU membrane began to decompose around 300 °C. When the temperature rose to 600 °C, the weight did not decrease significantly, and the final residual rate was 8.86%. Due to the low content of TPU in the PVDF/TPU composite membrane, the TG curve of the PVDF/TPU composite membrane is closer to that of PVDF. The initial decomposition temperature of the PVDF/TPU composite membrane is around 400 °C, and the final residual rate is 9.39%. Interestingly, the temperatures of the GO-PVDF/TPU and GO/Bi_2_S_3_-PVDF/TPU composite membranes began to decompose increasingly with the addition of GO and GO/Bi_2_S_3_. Obviously, the residue rate increased due to the presence of inorganic materials. This means that the GO-PVDF/TPU and GO/Bi_2_S_3_-PVDF/TPU composite membranes have better thermal stabilities than PVDF, TPU, and PVDF/TPU, which may be attributed to the GO or GO/Bi_2_S_3_ serving as nucleation agents and contributing to the increasing crystallization of PVDF [22]. Another reason for the improved thermal properties may be attributed to the covalent bonding between the GO or GO/Bi_2_S_3_ NPs and polymers [28]. Due to the same components of GO and GO/Bi_2_S_3_ in the film, the TG curves of the GO-PVDF/TPU and GO/Bi_2_S_3_-PVDF/TPU almost coincide. From the analysis of the thermal stability of the membranes, it can be concluded that the composite membranes with inorganic material can improve the thermal stability and are suitable for some medium temperature conditions.

### 2.3. Mechanical Property

In the process of production, cutting, and transportation, photothermal conversion materials are required to have some strength. Thus, the mechanical property of photothermal conversion material is an important parameter to evaluate the material properties. Figure 6 shows the mechanical properties of the PVDF, TPU, PVDF/TPU, GO-PVDF/TPU, and GO/Bi_2_S_3_-PVDF/TPU membranes, and the maximum strengths and elongations at break of the various membranes are listed in Appendix A. As shown in Figure 6, the stress and elongation-at-break of PVDF/TPU can be improved with the addition of TPU, owing to the existence of flexible polar crystals existing in TPU [29]. Moreover, with the respective introduction of GO and GO/Bi_2_S_3_, compared with the PVDF/TPU membrane, the mechanical performance of the GO-PVDF/TPU and GO/Bi_2_S_3_-PVDF/TPU membranes further evidently improved. This can be explained by the formation of hydrogen bonds or other covalent bonds between nanoparticles and polymer molecules, which makes the interface interaction strongly conducive to effective stress transfer in the process of stretching, thus improving the mechanical properties. In fact, GO has excellent mechanical properties and can improve the mechanical properties of the membrane [30,31].

### 2.4. Light Absorption Performance

When a semiconductor material is illuminated, electron holes pairs are created with the energy similar to the bandgap. The excited electrons eventually return to the lower energy level through a non-radiative relaxation and release heat as phonons, thereby establishing a temperature distribution based on light absorption and volume/surface composite properties [32,33]. Figure 7a shows the mechanisms of the light absorption and photothermal conversion. The solar absorption properties of different samples were evaluated by a UV-vis-NIR spectrometer, as shown in Figure 7c. Figure 7a,b displays the transmission and reflectance of the different membranes. As shown in Figure 7a, because of the small pore size and thin thickness, the transmissions of all the membranes are less than 20%. However, the membranes without inorganic additives have lower transmissions than GO-PVDF/TPU and GO/Bi_2_S_3_-PVDF/TPU, which have 2% and 1%, respectively. GO has a high reflectance due to its dependence on the incident light angle and high refraction, resulting in a higher reflectance of the GO-PVDF/TPU membrane (approximately 30%) than that of the GO/Bi_2_S_3_-PVDF/TPU membrane (approximately 5%) [4]. According to formula (2), GO/Bi_2_S_3_-PVDF/TPU has the highest light absorption (nearly 95%), followed by GO-PVDF/TPU, as shown in Figure 7d. This result can also be explained by the following principle. The multiple reflections and resonance scattering in the absorption materials help to reduce the reflection amount of the incident radiation so as to increase the solar absorption rate [34,35]. GO NPs and Bi_2_S_3_ NPs in GO/Bi_2_S_3_ composite NPs can interact with each other to increase the reflection and resonance scattering, thereby increasing the absorption rate of the membrane.

### 2.5. Photothermal Conversion Performance

The data of temperature changes (interval of 10 s) on different membranes surfaces are shown in Figure 8a,b. Figure 8c displays the infrared images of the GO/Bi_2_S_3_-PVDF/TPU membrane at every 30 s, with a total irradiation time of 300 s. In the absence of GO or GO/Bi_2_S_3_, the temperatures of PVDF, TPU, and PVDF/TPU membranes rose slowly and slightly; the equilibrium temperature of these membranes was around 50 °C, which was due to the poor solar harvestability. During the irradiation of 300 s, the temperatures of the GO-PVDF/TPU and GO/Bi_2_S_3_-PVDF/TPU membranes increased significantly and rapidly, and the equilibrium temperatures were 67 °C and 81 °C at the same irradiation time, respectively. In comparison with the GO-PVDF/TPU membrane, the equilibrium temperature of the GO/Bi_2_S_3_-PVDF/TPU membrane was further improved, indicating that the GO/Bi_2_S_3_ NPs have a better photothermal conversion performance than single-component GO, which benefits from the outstanding light absorbance of the Bi_2_S_3_ NPs. The temperature change in the GO/Bi_2_S_3_-PVDF/TPU membrane was significantly increased compared to other reports, as shown in Table 2 [6,7,8,36].

Stability is another important index by which to evaluate the performance of membrane applications. In order to study the stability of the PVDF, TPU, PVDF/TPU, GO-PVDF/TPU, and GO/Bi_2_S_3_-PVDF/TPU membranes, the temperature cycle tests of different membranes were carried out by radiation 300 s, naturally cooling for 300 s, and no irradiation subsequently. As shown in Figure 8a, the sample with the GO/Bi_2_S_3_-PVDF/TPU membrane heated up rapidly, which is attributed to the narrow bandgap and wide band absorption characteristics of Bi_2_S_3_. The final temperatures of the samples (PVDF, TPU, and PVDF/TPU membranes) were similar to the GO-PVDF/TPU and GO/Bi_2_S_3_-PVDF/TPU membranes, which is attributed to the low specific heat capacity of GO and Bi_2_S_3_ NPs. After five cycles of irradiation, the temperatures of the samples did not decrease significantly, and the initial temperature of each cycle increased slightly, which was caused by the increasing of the ambient temperature due to the irradiation time.

## 3. Materials and Methods

### 3.1. Materials

Graphene oxide (GO) aqueous solution was obtained from JCNANO, Inc. (Nanjing, China). Poly (vinylidene fluoride) (PVDF, *M_w_* = 1,100,000) and thermoplastic polyurethane (TPU, Mw = 600,000) were supplied by Solef Co. Ltd. (Shanghai, China) and Basf Co. Ltd. (Shanghai, China). Bismuth nitrate pentahydrate (Bi (NO_3_)_3_·5H_2_O) and polyvinylpyrrolidone (PVP) were provided by Nona technology Co. Ltd. (Hubei, China) and Zhengkun chemical Co. Ltd. (Tianjin, China). Thioacetamide (TAA) and ethylene glycol (EG) were obtained from Jiuding chemical Co. Ltd. (Shanghai, China). *N*,*N*-dimethylformamide (DMF), acetone, and ethanol were provided by Aladdin reagent Co., Ltd. (Shanghai, China).

### 3.2. Preparation of GO Nanoparticles

The GO nanoparticles (NPs) were prepared by adding 5 mL of GO aqueous solution and 5 mL of ethanol into a centrifuge tube (20 mL) via centrifugation. The centrifugation lasted until the mixture solution was layered, then we poured out the supernatant. An amount of 5 mL of ethanol was added to the above precipitate. The GO NPs were obtained by centrifugation and washing with ethanol several times to remove all the remnants, followed by drying at 120 °C for 2 h.

### 3.3. Synthesis of GO/Bi_2_S_3_

The GO-supported Bi_2_S_3_ (GO/Bi_2_S_3_) was prepared via a facile hydrothermal synthesis process (as shown in Figure 9a). In the typical synthetic route, 0.617 g of Bi(NO_3_)_3_·5H_2_O was dissolved in 5 mL of EG containing 0.640 g PVP and stirred at room temperature for 30 min until a transparent solution was formed. EG (50 mL) containing GO NPs (0.1 g) was added to the above mixture solution under stirring for 30 min and ultrasonic treatment for 10 min. Subsequently, TAA (0.207 g) was mixed with the above solution. The resulting mixture solution was transferred into a high-temperature and high-pressure reaction caldron with a Teflon liner and treated under 120 °C for 2 h. Finally, the target product GO/Bi_2_S_3_ NPs were obtained by naturally cooling, centrifugation, and washing with ethanol at least 3 times.

### 3.4. Fabrication of Nanofiber Membranes

Based on the compositions, PVDF, TPU, PVDF/TPU, GO-PVDF/TPU, and GO/Bi_2_S_3_-PVDF/TPU were labeled. The preparation methods of the different membranes are described in detail below. For clarity, the compositions of the different membranes are also available in Table 1. The nanofiber membranes were prepared using a self-made electrospinning device. The PVDF electrospun solution (11 wt%) was obtained by dissolving 0.11 g of PVDF powder into a mixture solvent (0.89 g) of DMF and acetone (at the mass ratio of 70:30) with magnetic stirring at 50 °C for 2 h. The spinning parameters for the PVDF nanofibers were as follows: the voltage of 15 kV, needle-to-collector distance of 20 cm, feeding rate of 1 mL/h. The electropinning solutions of different concentrations of PVDF (10 wt% and 12 wt%) were prepared in accordance with the above methods. The morphology of the PVDF nanofibers was compared with different concentrations. The PVDF nanofibers with a better morphology were obtained when the concentration of electrospinning solution was determined to be 11 wt% (as shown in Appendix A).

The TPU electrospun solution (24 wt%) was obtained by dissolving TPU particles of 0.24 g into a mixed solvent of DMF and acetone (0.76 g, at the mass ratio of 70:30) with magnetic stirring for 1 h at 60 °C. The applied voltage was 35 kV, the distance between the needle and collector was 38 cm, and the feeding rate was 1.2 mL/h. TPU electrospinning solutions with concentrations of 23 wt% and 25 wt% were also prepared. By comparing the morphology of TPU nanofibers, it can be concluded that the fiber morphology is better when the concentration of the electrospinning solution is 24 wt% (as shown in Appendix A).

The PVDF/TPU electrospun solution was obtained through blending the PVDF solution and a TPU solution at the mass ratio of 6:4 for 3 h. TPU/PVDF composite nanofiber membrane was a fabrication with the voltage of 25 kV and feed rate of 1 mL/h. During the PVDF/TPU electrospinning process, the distance between the needle and the receiving roller was 30 cm. The mass ratio of PVDF to TPU is determined by the average diameter and CV value of the fibers prepared in different proportions (as shown in Appendix A).

GO-PVDF/TPU and GO/Bi_2_S_3_ -PVDF/TPU electrospun solutions were obtained through adding GO and GO/Bi_2_S_3_ (10 wt%, based on the total mass of PVDF and TPU) into the mixture solution of PVDF/TPU, respectively (as shown in Figure 9b). For successful electrospinning, the spinning solution was treated by ultrasound at room temperature for 2 h. The concentration of nanoparticles is also determined by the fiber morphology. When the concentration of nanoparticles exceeds 10 wt%, the aggregation phenomenon is obvious, resulting in uneven fiber thickness and fusiform structure, as shown in Appendix A. During the preparation of the GO-TPU/PVDF and GO/Bi_2_S_3_-TPU/PVDF electrospinning processes, the voltage, feed rate, needle specification, and distance between the needle and receive roller were the same as with the TPU/PVDF membrane.

The temperature and relative humidity of all the electrospinning process was 27 ± 2 °C and 30% ± 2%, respectively. All the needles used in the spinning process were gauge 20.

### 3.5. Characterization

The elemental composition and chemical information of the GO/Bi_2_S_3_ NPs were analyzed by x-ray photoelectron spectroscopy with an excitation source of Al Kα radiation (XPS, Thermo ESCALAB 250XI, Waltham, USA). The crystallographic structure of the GO/Bi_2_S_3_ NPs was analyzed by using powder x-ray diffraction patterns (XRD, Ultima IV, Tokyo, Japan) at a scanning rate of 5° min^−1^ in the 2θ range of 5°–90°. A transmission electron microscope (TEM, JEOL JEM 2100F, Tokyo, Japan) was used to observe and analyze the morphology, particle size, dispersion, and crystal structure of the GO/Bi_2_S_3_ NPs. The morphologies of the nanofiber samples were evaluated by field emission scanning electron microscopy (FESEM, S-4800, Hitachi, Tokyo, Japan). The diameter of the nanofibers was measured using the Image Pro software. The preparation of the GO-PVDF/TPU and GO/Bi_2_S_3_-PVDF/TPU nanofibers was further confirmed by Fourier transform infrared spectroscopy (FTIR, JKY/BTR111 Infrared Raman spectrometer, Kolegwani (Shanghai) Analytical Instruments Co., LTD, Shanghai, China) in the range of 400–4000 cm^−1^. The capillary flow porometer (CFP-1100AX, PMI Corporation, Shanghai, China) was used to measure the pore size of the membranes. The porosity of membrane was obtained by immersing the specimens (2 cm circle in diameter) into n-butanol for 2 h and based on the Equation (1):(1)$$\;\mathrm{Porosity}\left(\%\right)=\frac{\left(m_{w}-m_{d}\right)/\rho }{m_{w}/\rho +sd}$$
where *m_d_* and *m_w_* represent the mass before and after soaking the n-butanol, respectively. *ρ* is the density of the n-butanol. *s* and *d* are the area and thickness of the sample.

The mechanical property of the as-prepared membrane was tested by an Instron 3369 Universal Strength Tester at a speed of 20 mm·s^−1^, with the specimen dimension about 2 × 10 cm^2^. The thermal stability of the membrane was evaluated by differential scanning calorimetry (DSC, Netzsch DSC 214). The measurement was performed at 30–350 °C with a heating rate of 10 °C·min^−1^ under a flow of 75 mL·min^−1^ high purity nitrogen. A thermogravimetric analysis (DTG-60H, Shimadzu, Tsushima, Japan) was performed to measure the thermogravimetric curves of the various samples. The study was performed under a flow of 75 mL·min^−1^ high purity nitrogen at 30–800 °C, with a heating rate of 20 °C·min^−1^. The reflectance (R) and transmittance (T) of the nanofiber membrane was recorded by the ultraviolet-visible-near-infrared (UV-Vis-NIR) spectrometer (UV 3600, Shimadzu, Tsushima, Japan) over the full wavelength. The absorption (A) was calculated according to Equation (2).
(2)A%=1−R%−T%

### 3.6. Photothermal Conversion Test

To investigate the photothermal conversion property of the membranes, circular samples with a diameter of 30 mm were added into a polypropylene (PP) container and irradiated by a xenon lamp (HGILX500, 500 w, Yunke instrument, China) with a light intensity of 1 kW·m^−2^ [8] at the temperature of 27 °C and relative humidity of 35%. The temperatures of the samples and infrared image were obtained by the infrared thermography (Tis10, Fluke, Everfried, WA, USA). In order to ensure the accuracy of the temperature test, the surface temperature of each sample center was measured. The illumination stability of the different membranes was determined by cyclic testing. They were irradiated for 300 s firstly and cooled naturally for 300 s subsequently in each cycle.

## 4. Conclusions

In conclusion, we designed and fabricated a GO/Bi_2_S_3_-PVDF/TPU composite nanofiber membrane with prominent light absorption and a photothermal conversion property. Due to the introduction of the Bi_2_S_3_ NPs, the absorbance and temperature changes of the GO/Bi_2_S_3_-PVDF/TPU composite membrane were markedly improved. The GO/Bi_2_S_3_-PVDF/TPU composite membrane has the highest light absorption rate (about 95%) in the wavelength range of 400–2500 nm. After 300 s of solar irradiation, the GO/Bi_2_S_3_-PVDF/TPU membrane temperature can reach as high as 81 °C, while the final equilibrium temperature is only about 50 °C for the PVDF, TPU, and PVDF/TPU membranes. The result means that the photothermal conversion performance will be significantly improved. Moreover, because of the addition of TPU, the mechanical property of the GO/Bi_2_S_3_-PVDF/TPU membrane was significantly ameliorated. The existence of Bi_2_S_3_ NPs also contributed to improving the mechanical properties and thermal stability. The improvement of the mechanical property and thermal stability means that the composite membrane has a longer service life and more applications. In a word, our results shed light on a promising use of a GO/Bi_2_S_3_-PVDF/TPU composite nanofiber membrane as a photothermal conversion material.

## Figures and Tables

**Figure 1 ijms-21-04224-f001:**
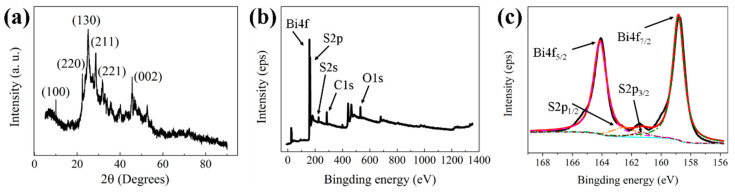
(**a**) XRD spectra of the graphene oxide (GO)/Bi_2_S_3_ NPs, (**b**) XPS survey spectra of the GO/Bi_2_S_3_ nanoparticles (NPs) and (**c**) the high-resolution spectra of Bi4f and S2p.

**Figure 2 ijms-21-04224-f002:**
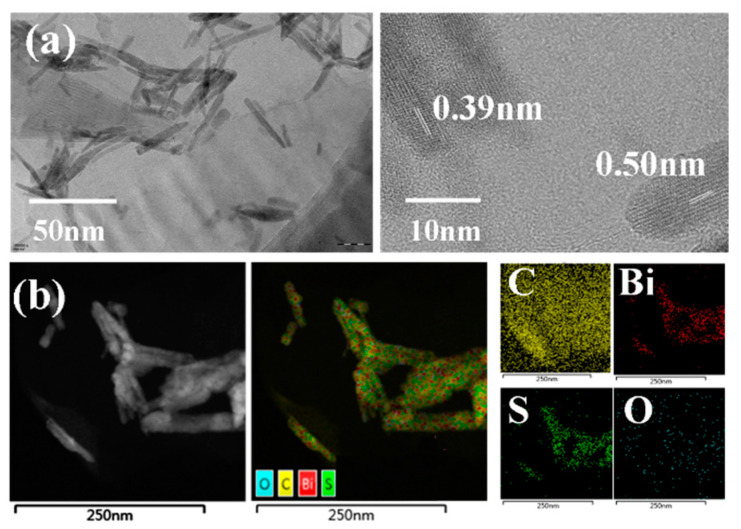
(**a**) TEM and HRTEM images of the GO/Bi_2_S_3_ NPs; (**b**) HAADF and EDS mapping.

**Figure 3 ijms-21-04224-f003:**
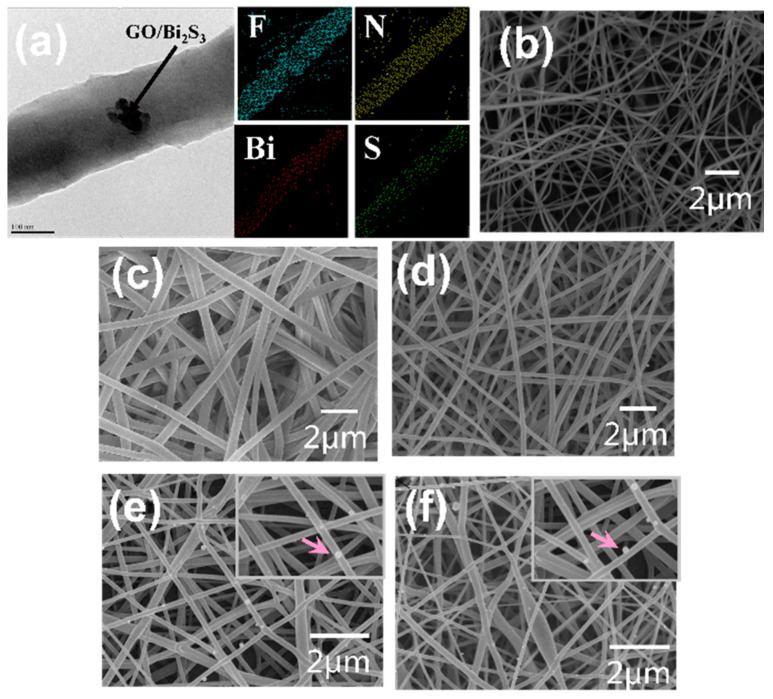
(**a**) The TEM image of GO/Bi_2_S_3_-PVDF/TPU and elements mapping of the F, N, Bi, and S. SEM images of (**b**) PVDF, (**c**) TPU, (**d**) PVDF/TPU, (**e**) GO-PVDF/TPU and (**f**) GO/Bi_2_S_3_-PVDF/TPU nanofibers.

**Figure 4 ijms-21-04224-f004:**
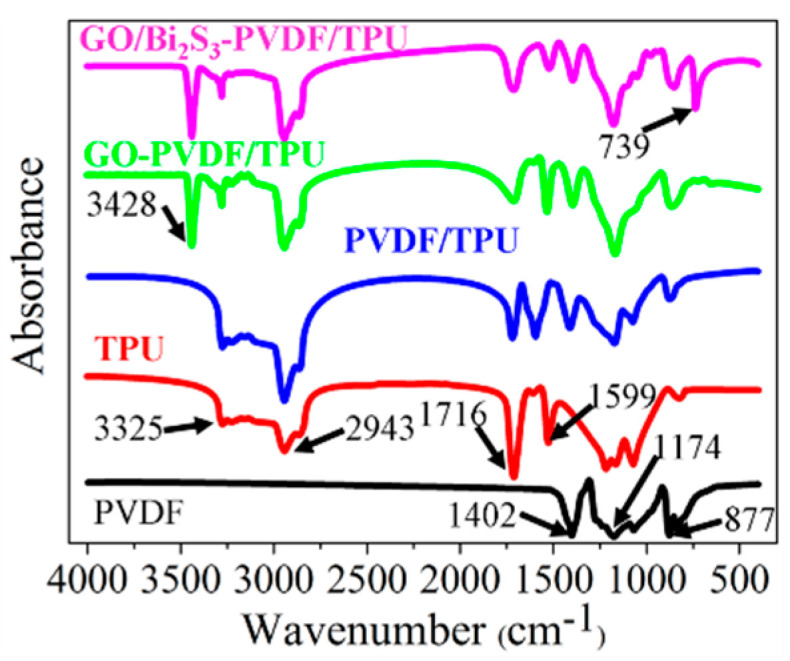
FTIR of different membranes.

**Figure 5 ijms-21-04224-f005:**
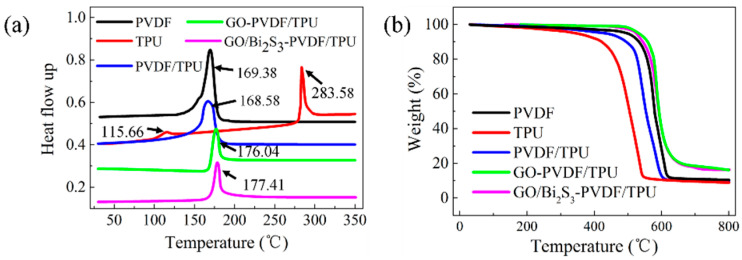
Differential scanning calorimetry (DSC) (**a**) and TG (**b**) of different membranes.

**Figure 6 ijms-21-04224-f006:**
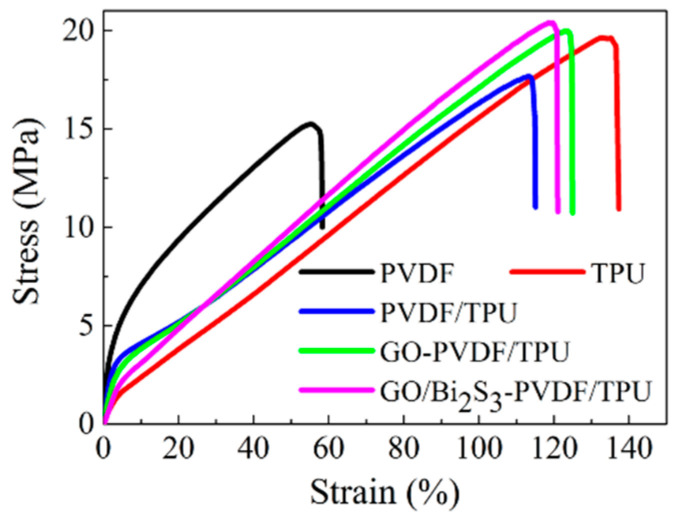
Stress–strain curve of different membranes.

**Figure 7 ijms-21-04224-f007:**
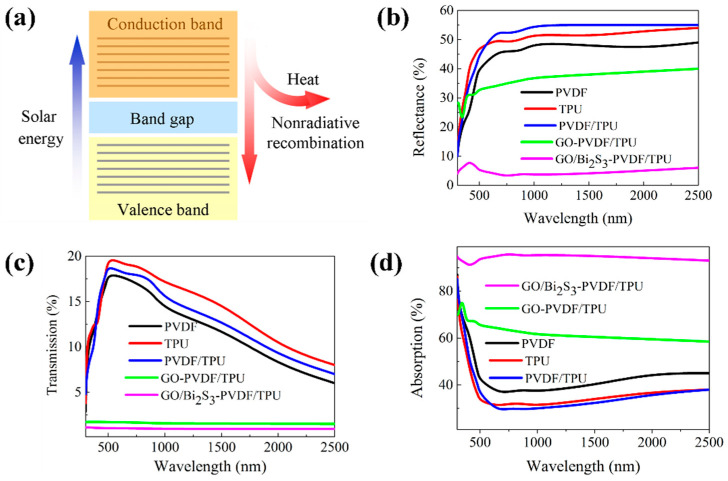
The mechanisms of light absorption and photothermal conversion (**a**), UV-Vis-NIR transmittance (**b**), and reflectance (**c**) and absorption (**d**) spectra of different membranes.

**Figure 8 ijms-21-04224-f008:**
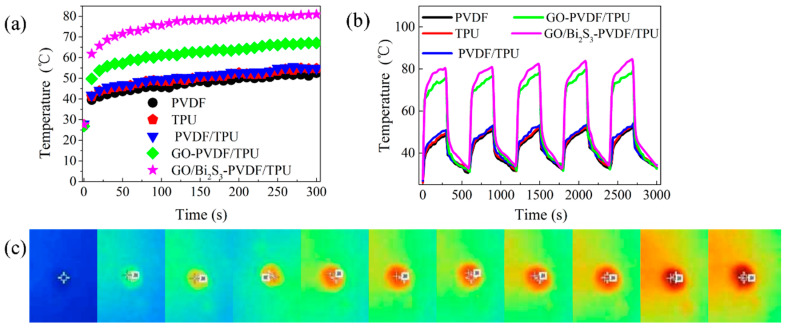
(**a**) Temperatures on the surface of different membranes under irradiation. (**b**) Temperature of five cycles. (**c**) Infrared images of the GO/Bi_2_S_3_-PVDF/TPU membrane under irradiation.

**Figure 9 ijms-21-04224-f009:**
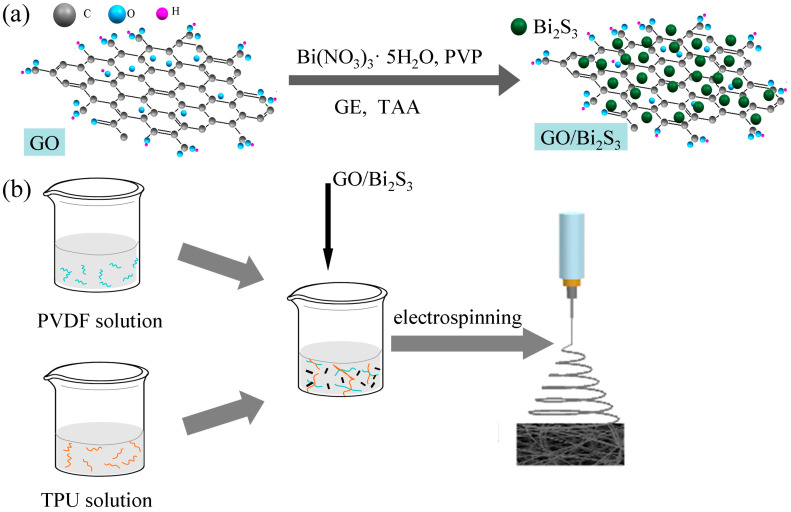
Schematic illustration of the synthesis of GO/Bi_2_S_3_ NPs (**a**) and preparation of the GO/Bi_2_S_3_-PVDF/TPU nanofibers membrane (**b**).

**Table 1 ijms-21-04224-t001:** Compositions and structural parameters of the samples.

Membrane	Compositions	Average Fiber Diameter (nm)	Average Pore Size (ìm)	Porosity (%)
PVDF	Pristine PVDF electrospun nanofiber	152	1.8	70
TPU	Pristine TPU electrospun nanofiber	425	2.3	65
PVDF/TPU	Nanofibers were spun by PVDF and TPU mixed solution	240	2.1	73
GO-PVDF/TPU	PVDF/TPU electrospun nanofibers loaded with GO NPs	149	1.3	83
GO/Bi_2_S_3_-PVDF/TPU	PVDF/TPU electrospun nanofibers loaded with GO/Bi_2_S_3_ NPs	129	1.2	73

**Table 2 ijms-21-04224-t002:** Temperature changes in various membranes under the same irradiation intensity.

Samples	Irradiation Time (s)	Temperature Change (°C)
PDA/PEI/PPy@PI nanofiber membrane	3600	11.9
CNT-silica bilayered material	400	50
h-G foam	300	33.5
TiO_2_-PDA/PPy/cotton	600	47.7
GO/Bi_2_S_3_-PVDF/TPU	300	53.3

PDA/PEI/PPy@PI nanofiber is composed of hydrophobic PI layer (substrate), photothermal PPy layer (intermediate layer) and hydrophilic PDA/PEI layer (outlayer). CNT-silica bilayered material consists of a hydrophobic CNT film (top layer) and a macroporous silica substrate (bottom layer). Hierarchical graphene foam is defined as h-G foam. CNT-silica bilayered material. The TiO_2_-PDA/PPy/cotton material is obtained by in-situ polymerization of Py on cotton fabric and deposition of TiO_2_ nanoparticles.

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
