# Peer review of "GO/Bi2S3 Doped PVDF/TPU Nanofiber Membrane with Enhanced Photothermal Performance"

_ijms, 2020, doi:10.3390/ijms21124224_

Round 1

Reviewer 1 Report

This work as a proposal for a solution of solar distillation raises my great doubts. In the current state, the authors have not shown whether there is a real chance to stop their solution.Hence, the work should be supplemented with further research or based on current research write it in a completely different concept.

The intensity of solar radiation reaching the upper atmosphere limits is determined by solar radiation. This value depends on the average Earth-Sun distance and is about 1366.1 W / m². There is even less to the surface of the installation.

Evaporation of 1 kg of water requires 2400 kJ / kg. Hence the yields are not high.To avoid salt crystallization on the membrane surface, the solution must be pumped, which requires electricity.To have some reasonable efficiency of desalinated water, a very large membrane surface should be mounted. The authors should make a summary of the expected costs of membrane for such a large installation area and the costs of pumping recirculated water in such a large installation.

L58

Solar radiation is very aggressive to polymers. One of the studies showed that after 70 hours of exposure the membranes were scattered into powders. Therefore, without testing the changes in mechanical strength after 100 hours of exposure, all this work becomes only a description of the chemical processes in the technological proposal, most likely having no chance for practical application.

If such tests do not prove suitability for solar distillation, then this work should be re-written and only polymer issues should be considered.Distilled water as a feed also adds little. Please supply the installation with 35 g / L NaCl solution.

Author Response

Dear Reviewer

We are truly grateful to your critical comments and thoughtful suggestions. Based on these comments and suggestions, we have made careful modifications in the manuscript. All revised portion are marked in highlight in the revised paper. We hope the improved manuscript will meet your standard. If there are still problems with our work, please don’t hesitate to tell us. Below you will find our responses to your comments:

  1. This work as a proposal for a solution of solar distillation raises my great doubts. In the current state, the authors have not shown whether there is a real chance to stop their solution. Hence, the work should be supplemented with further research or based on current research write it in a completely different concept.

Answer: As the suggestion of the reviewer, we changed the focus of the article based on the current research. The revised paper focuses on describing the light absorption properties and photothermal conversion properties of the GO/Bi2S3-PVDF/TPU composites rather than the water evaporation applications. All descriptions, tests and results analysis of water evaporation have been removed. The application of photothermal conversion materials is added in the introduction part (L40 P1). The test method of photothermal conversion performance is rearranged (L183 P5). In the abstract and conclusion part, the properties of light absorption and photothermal conversion property are also described (L28 P1, L367 P12).

Introduction part

Photothermal conversion is an emerging approach of using light energy, which can be widely used in the fields of heat therapy, domestic water heating, photocatalysis and other field[1-3].

2.6 photothermal conversion test

To investigate the photothermal conversion property of the membranes, the circular samples with a diameter of 30 mm were added into a polypropylene (PP) container and irradiated by a xenon lamp (HGILX500, 500w, Yunke instrument, China) with a light intensity of 1kW·m-2[8],at the temperature of 27 and relative humidity of 35 %. The temperature of the samples and infrared image were obtained by the infrared thermography (Tis10, Fluke, USA). In order to ensure the accuracy of the temperature test, the surface temperature of each sample center was measured. The illumination stability of the different membranes was determined by cyclic testing. Irradiated for 300 s firstly and cooled naturally for 300 s subsequently in each cycle.

Abstract

Finally, the results show that, the GO/Bi2S3-PVDF/TPU composite membrane has the highest light absorption rate (about 95%) in the wavelength range of 400nm-2500nm. In the 300 s irradiation process, the temperature changes of the GO/Bi2S3-PVDF/TPU composite membrane was the most significant and rapid, and the equilibrium temperature of the same irradiation time was 81.

Conclusion

The GO/Bi2S3-PVDF/TPU composite membrane has the highest light absorption rate (about 95 %) in the wavelength range of 400 -2500nm.  After 300s solar irradiation, the GO/Bi2S3-PVDF/TPU membrane temperature can reach as high as 81 , while the final equilibrium temperature is only about 50 for the PVDF, TPU and PVDF/TPU membrane. The result means that the photothermal conversion performance will be significantly improved.

Reviewer 2 Report

The authors have sufficiently addressed the reviewers' comments and the current revised manuscript can now be recommended for acceptance.

Author Response

Thank you for your valuable comments and recognition

Round 2

Reviewer 1 Report

  After removing desalination, the article describes
what the authors know and sepkulations have been eliminated.
That is why such a correction can be published.

This manuscript is a resubmission of an earlier submission. The following is a list of the peer review reports and author responses from that submission.

Round 1

Reviewer 1 Report

The subject of the work is related to the process of so-called solar distillation, which is known and implemented industrially nearly 100 years ago. The advantage is getting fresh water in places where there is no other drinking water. However, despite many attempts, its basic disadvantage, i.e. very low efficiency in relation to the occupied area, was not overcome. Also the authors' proposal does not solve this problem.

The main part of the work is the production of new material with interesting properties. This issue could be the subject of a work describing it.

However, the current proposal as a water desalination method has little chance of industrial start-up, not only because of the low efficiency but also because of huge process problems, which the authors do not even mention - and they should get a little familiar with it since they propose an article in this topic.

The low efficiency is also due to the fact that the energy supplied by solar radiation per 1 m2 of surface is very small compared to the energy needed to evaporate water - over 2500 kJ / L.

In the authors' solution, the capillaries should be laid out in a single layer, one next to the other, which will occupy a large area and we will obtain a small evaporation surface. In this way, the advantages of membrane capillary modules - a large working area in a small volume - will not be obtained. In this case, another solution - photovoltaic cells that produce electricity driving the reverse osmosis system is much more efficient.

Even if we do this, we do not evaporate distilled water, but water containing salt. As a result, salt quickly crystallizes and destroys such membranes proposed by the authors. You can't propose such technologies without solving the scaling problem.

Evaporation does not end the process - and how do the authors want to condense water?

Reviewer 2 Report

  1. Polish the English throughout. There are a number of grammar and structure issues. Proofread many times. Also, for technical papers, this phrase “the only fly in the ointment” is not appropriate.

        -2.2.:”Preparation” of GO…

  1. P3, L105-106: Explain why is there a need for a mechanical requirement for solar absorbers? They are usually used as floating structures and not made into modules.
  2. 2.2: Do not start a sentence with a number à 5 ml
  3. Give the properties and sizes of your GO and Bi2S3. These are important details.
  4. Present in a table the different prepared samples.
  5. Why did you choose 10 wt% fo Bi2S3/GO? Based from my experience, this is a very high concentration that leads to clogging of the spinneret.
  6. Did you do optimisation of your electrospinning parameters including the concentration of your nanoparticles?
  7. P5, L193-195: What is the basis of your cotton fabric design with 8 strips?
  8. P5, L198-199: How did you measure the temperature? Whish parts did you measure? Provide more details.
  9. Mention in the experimental methods the light absorption tests.
  10. From Fig 6, how much is solar absorptivity of your composite nanofiber? Mention the actual value in the explanation.
  11. From Fig 2, it is clear that the nanoparticles are embedded inside the polymeric nanofiber and are not uniformly distributed. How can you justify the increase in temperature during solar irradiation (Fig. 7) when most particles are embedded in the polymer? Polymers have low thermal conductivity. Explain the mechanism of solar evaporation using your material (show schematic).
  12. Most of the details in the discussion are just a description of the results but no in-depth analysis and discussion. This has to be improved more.
  13. P9, L334: I can’t see evidence of the uniform distribution in GO NPs.
  14. Any result for the solar-thermal conversion performance?
  15. Experimental design and depth of analysis need to be improved.